# Rare Onset of Tubercular Peritonitis Amidst Chronic Renal Dysfunction

**DOI:** 10.3390/reports6040044

**Published:** 2023-09-22

**Authors:** Romeo Popa, Cristian-Corneliu Georgescu, Daniel-Cosmin Caragea, Daniela Cana-Ruiu, Cristina Ene, Lucretiu Radu, Victor Gheorman, Marius-Ciprian Varut, Veronica Gheorman, Andrei Orezanu, Andrei Razvan Codea, Mirela Ghilusi, Loredana-Adriana Popa, Magdalena Diaconu

**Affiliations:** 1Department of Pharmacology, University of Medicine and Pharmacy of Craiova, 200349 Craiova, Romania; romeo_rop@yahoo.com (R.P.); cgeorgescu2001@yahoo.com (C.-C.G.); cristina.ene01@yahoo.com (C.E.); diaconumagda@yahoo.com (M.D.); 2Department of Nephrology, University of Medicine and Pharmacy of Craiova, 200349 Craiova, Romania; caragea.daniel87@yahoo.com (D.-C.C.); daniela_cana@yahoo.com (D.C.-R.); 3Department of Hygiene, Faculty of Nursing and Midwives, University of Medicine and Pharmacy of Craiova, 200349 Craiova, Romania; 4Department of Psychiatry, University of Medicine and Pharmacy of Craiova, 200349 Craiova, Romania; victor.gheorman@umfcv.ro; 5Department of Biophysics, University of Medicine and Pharmacy of Craiova, 200349 Craiova, Romania; varutmarius@yahoo.com; 6Department of Cardiology, University of Medicine and Pharmacy of Craiova, 200349 Craiova, Romania; 7Department of Urology, Fundeni Clinical Institute, 022328 Bucharest, Romania; drorezanu@gmail.com; 8Department of Internal Medicine, University of Agricultural Sciences and Veterinary Medicine, 400370 Cluj-Napoca, Romania; razvancodea@yahoo.com; 9Department of Pathological Anatomy, University of Medicine and Pharmacy of Craiova, 200349 Craiova, Romania; mirela@dass.ro; 10Faculty of Dentistry, University of Medicine and Pharmacy of Craiova, 200349 Craiova, Romania; adriana_lautaru@yahoo.com

**Keywords:** chronic kidney disease, peritoneal tuberculosis, exploratory laparotomy

## Abstract

Tuberculosis Peritonitis is a serious condition, whose diagnosis is established late due to the nonspecific nature of the clinical features, which delays the performance of imaging investigations and, implicitly, the setting of the diagnosis through biopsy and histopathological examination. We report the case of a 49-year-old man who presented in our clinic with nonspecific symptoms and significant nitrogen retention, with ascites fluid detected during the clinical–paraclinical examination, ultimately confirming the diagnosis of bacillary peritonitis. Confirmation of tuberculous etiology through biopsy and/or bacteriological examination is sovereign for the diagnosis. The therapeutic protocol includes three anti-tuberculostatic drugs, for a period of at least 6 months, with or without the combination of corticosteroid therapy during the first months of treatment. The patient evolution under treatment was initially favorable, but due to peritoneal adhesions, it underwent complications later.

## 1. Introduction

Tuberculosis (TB) is a pressing global public health issue, contributing significantly to both morbidity and mortality. Prior to the emergence of the COVID-19 pandemic, TB held a more prominent position than HIV/AIDS in terms of its impact on mortality [1], resulting in about 1.3 million annual fatalities and 10 million new cases in 2018, with over half occurring in developing nations [2]. While the prevalence and mortality of pulmonary TB has decreased globally due to anti-tuberculosis drugs and socio-economic improvements, the opposite trend has occurred with extra-pulmonary TB [3].

Lung involvement is common in TB, with about 12.5% being extra-pulmonary cases. Abdominal TB accounts for 11–16% of extra-pulmonary cases, affecting the bowel, peritoneum, and lymph nodes. Timely diagnosis of tuberculous peritonitis is vital, as delayed treatment increases mortality risk. Intraperitoneal tuberculous abscess, an uncommon and serious form of extra-pulmonary TB, necessitates early and accurate diagnosis for effective treatment [4].

Tuberculous peritonitis (TP) is an infrequent form of *Mycobacterium tuberculosis* extra-pulmonary infection, constituting up to 3.5% of TB cases and 31–58% of abdominal TB cases [5]. BP risk escalates in patients with underlying conditions, like cirrhosis, HIV/AIDS, neoplasms, and diabetes. Elevated risk is also seen in those on corticosteroids, cytotoxic, immunosuppressive, and immunomodulator drugs, along with individuals with chronic kidney disease (CKD), including hemodialysis (HD), peritoneal dialysis, and kidney transplant recipients [5,6]. Infection typically arises from reactivated latent peritoneal tuberculosis, often with hematogenous dissemination from a primary pulmonary source or via lymphatic spread. Infrequently, mycobacteria infiltrate the transmural peritoneal cavity, as seen in small intestine infection, or through the ingestion of tuberculous mycobacteria, such as unpasteurized milk consumption [7].

In a meta-analysis conducted by Ayinalem A. and collaborators, it was noted that the incidence of TB in patients with CKD ranges from 60 per 100,000 in the United Kingdom to 19,270 per 100,000 in China. In particular, regarding TP incidence, patients undergoing hemodialysis (5611 per 100,000) and peritoneal dialysis (3533 per 100,000) exhibited a higher incidence of TB compared to patients with renal transplantation (2700 per 100,000) and pre-dialysis CKD (913 per 100,000) [8].

Diagnosing this condition is challenging due to lacking distinct clinical signs, low specificity in common diagnostic tests, and nonspecific ultrasonographic and radiological findings. Isolating mycobacteria from ascitic fluid is often tricky, as it is usually paucibacillary. Consequently, peritoneal biopsy stands as the gold standard [9]. Before tuberculostatic treatment became available, the prognosis in BP was unfavorable, with a mortality rate of approximately 50% of cases [10].

The purpose of the research titled “Rare Onset of Tubercular Peritonitis Amidst Chronic Renal Dysfunction” is to investigate and enhance the understanding of the unique clinical scenario where tubercular peritonitis exacerbates CKD.

Despite the negative result for BK obtained in the biochemical examination, the positive ADA (adenosine deaminase) reaction and the presence of frequent lymphocytes in the cytological examination prompted us to conduct histopathological tests. In the end, based on histopathological test, we had strong evidence of Mycobacterium Tuberculosis in ascitic fluid.

## 2. Detailed Case Description

The study protocol was approved by the Ethics Committee and is in accordance with the Declaration of Helsinki of 1975. Subjects gave their informed consent for inclusion before participating in the study. In our research, the process involved explaining the purpose of the study, procedures, potential risks and benefits, and emphasized voluntary participation. We ensured that the participants fully understood the information provided, their rights in the study, and the confidentiality measures. Informed consent involved verbal discussions and electronic consent forms.

This paper presents the case of a 49-year-old male patient from an urban environment, who presented to the Nephrology department with marked physical asthenia, nausea, lack of appetite, pain in the lower limbs, dry cough, headache, dizziness, intense dehydration of the skin and mucus, and pain in the left flank.

The medical history included kidney stones diagnosed in 2005, complicated with bilateral ureter-hydronephrosis (UHN) requiring retrograde ureteroscopy and in situ lithotripsy with bilateral ureteral catheter, parathyroidectomy diagnosed in 2013 for parathyroid adenoma, high blood pressure values from 2013 (220/110 mmHg), and CKD from 2005 with repeated exacerbations treated by intermittent hemodialysis.

The clinical examination of the patient shows an influenced general condition, fever (39 °C), BMI-15 kg/m^2^, cutaneous and mucosal dehydration, cardiovascular and pulmonary balanced, with canker sores on uvula. The abdomen was slightly sensitive to palpation in the left quadrant and displaceable dullness in the left quadrant on percussion, presenting a low movement of bowels in the last 2 weeks and palpable spleen under the left ribcage (Figure 1).

The biological balance reveals anemia (Hb 7.3 g/dL), inflammation (C reactive protein 12 mg/L, fibrinogen 497 mg/dL, procalcitonin 1.2 ng/dL), acidosis, and important nitrogen retention (creatinine 8.3 mg/dL, urea 205 mg/dL, glomerular filtration rate eGFR = 7 mL/min, 1.73 m^2^). Urine samples showed moderate proteinuria (1404 mg/24 h) abnormal urinary concentration. Notably, urine cultures, liver virus tests, and HIV tests all returned negative results. (Table 1.)

EKG: Sinus rhythm, rate instead = 90 bpm without repolarization phase changes. Cardiopulmonary radiography ruled out active pleuro-pulmonary changes. Abdominal X-ray showed moderate bloating, middle abdominal floor, and left hypochondrium.

Abdominal–pelvic ultrasound reveals undersized kidneys (7 cm in long axis bilaterally) with a completely disorganized structure, UHN grade II on the left kidney, a splenomegaly with dilatations at the level of the hilum, and a small amount of ascitic fluid.

Native tomographic examination reveals perihepatic, perisplenic, paracolic abdominal fluid in large quantities, and enlarged spleen with venous dilatations in the hilum.

The cardiac ultrasound did not show any structural or functional abnormalities. In order to clarify the etiological diagnosis of ascites, an abdominal and pelvic MRI examination was performed, without providing additional information, except for the presence of a large amount of perihepatic, perisplenic, and paracolic abdominal fluid.

Given the altered general state, the presence of febrile syndrome, and the presence of ascites fluid in large quantities on examination of native MRI of the abdomen and pelvis, with exudate nature in a patient with multiple signs of secondary CKD immunosuppression, the suspicion of TP diagnosis was raised, thus leading to the decision to perform paracentesis in order to establish the etiology of ascites fluid.

Performing paracentesis confirmed the presence of peritoneal exudate, with initial negative cultures, which raised the suspicion of TP. The ascitic fluid smear, stained with Ziehl–Nielsen dye, underwent a direct examination for BK, which was was negative, and the adenosine deaminase reaction (ADA reaction) was positive (42 U/L). The high level of glucose determined in the ascites fluid (85 mg/dL) could be a consequence of other associated pathologies. The typical level of glucose in TB ascitic fluid is around 50 mg/dL or even lower. The presence of leukocyte and neutrophil cells in the ascites fluid indicates bacterial infection, with the cultures being positive for *Klebsiella* and Enterobacter. Also, the high level of lymphocytes indicates the presence of BP. The cytological examination reveals isolated mesothelial cells, common lymphocytes, leukocytes, PMN, neutrophils, and rare red blood cells (Table 2), thus leading to the decision to perform peritoneal biopsy.

Through laparoscopy, the tubercular nodule is observed macroscopically, and the histopathological results from hematoxylin-and-eosin staining indicate the presence of epithelial granuloma that forms Langhans cells that could explain the presence of a chronic inflammatory reaction (Figure 2).

Antituberculosis therapy was initiated with isoniazid, rifampicin, and pyrazinamide for six months, with doses adjusted for renal function according to guidelines and specific specialty protocols, and antibiotics (beta-lactamase with Meropenem) for concurrent infection with Enterobacter and *Klebsiella* Spp. Pathogenic therapy was performed with dexamethasone.

Initially, the patient’s clinical condition improved and digestive tolerance was restored after treatment, with improved nitrogen retention (creatinine 3.3 mg/dL eGFR = 21 mL/min/1.73 m^2^). Subsequently, intestinal bowel disorders reappeared: initially, food vomiting reduced in quantity, there was the absence of bowel movement for a few days, and then diarrhea after 5–7 days.

After this episode, there was a significant amount of fecal vomiting and diarrhea. A simple abdominal X-ray was performed, where important hydro-aerial levels were highlighted, the reason being that the diagnosis of intestinal occlusion was formulated, and the patient was transferred to the surgery department (Figure 3.).

As a result of the intensification of the important nitrogen retention (creatinine = 8 mg/dL, urea = 300 mg/dL, eGFR = 4 mL/min/1.73 m^2^), but also in order to reduce the anesthetic risk, HD was initiated. Intraoperative: the adhesive band near the ileocecal valve showed complete intestinal obstruction with large phytobezoar. Postoperatively, the patient evolution was unfavorable, with patient death occurring 7 days after surgery.

Corroborating the clinical and paraclinical data, the most important diagnoses were as follows: bacillary and bacterial peritonitis (*Klebsiella* Spp., Enterobacter). Other secondary diagnostics could be formulated as follows: acute CKD, chronic tubulointerstitial nephropathy, UHN left grade II, severe metabolic acidosis, secondary hyperparathyroidism, severe hypocalcemia, severe hyperphosphatemia, and severe hyperuricemia. Mixed dyslipidemia and secondary anemia were also present.

## 3. Discussion

Bacteriological and cytological ascites fluid evaluation is crucial for diagnosis. Elevated PMN levels aid in distinguishing uncomplicated ascites from spontaneous bacterial peritonitis (SBP); a PMN count > 250/mm^3^ strongly suggests fluid infection, though it is also seen in malignant ascites. Abundant lymphocytes and negative bacteriological tests hint at BP or peritoneal carcinomatosis, although lymphocyte absence in CKD patients remains unexplained in some studies [10].

Adenosine deaminase (ADA) levels are a newer ascitic molecular diagnostic tool to detect mycobacteria [11]. ADA levels indicate tuberculosis by reflecting T-lymphocyte stimulation via mycobacterial antigens. ADA’s diagnostic utility in ascites fluid for peritoneal tuberculosis varies in the literature: Riquelme et al. reported high specificity (97%) and sensitivity (100%) using Giusti’s method [12]. Voigt M.D et al. found ADA less specific, showing lower sensitivity in cirrhosis, false positives in bacterial peritonitis, and malignancies. In high-TB incidence settings or high-risk patients, ADA in ascites may screen, yet this is not a reliable biomarker [13,14]. In cases where lymphocytes predominate in effusions, the most common threshold for an ADA test result indicating tuberculosis (TB) is an ADA level greater than 40 U/L. An ADA level greater than 40 U/L exhibits sensitivity ranging from 87% to 93% and specificity ranging from 89% to 97% for TB. At a higher threshold of 35 U/L, the sensitivity is higher (93–95%) but specificity is lower (74–90%). [15]. Diagnosis and proper treatment still necessitate peritoneal exploration and examination [16].

Laparoscopy allows for inspection of the peritoneum as well as the option of pathological and microbiological confirmation of the diagnosis. Laparoscopy with biopsy confirms tuberculous peritonitis in 85 to 90% of cases [17].

In this case report, co-infection of ascites fluid with *Klebsiella* and Enterobacter likely resulted from echo-guided paracentesis, a rare complication (around 2.46%) noted in a retrospective study of 1218 patients with 4389 paracenteses [18]. Despite improved imaging (CT/MRI, ultrasound), exploratory laparoscopy with biopsy remains the gold standard for diagnosing ascites fluid’s etiology [19]. Epithelial granuloma and Langhans cell type with mesothelial hyperplasia strongly support TP diagnosis [20].

Our findings regarding the importance of laparoscopy in BP also support previous work in the domain, which stipulates that laparoscopy is the most specific diagnostic test for BP. The main advantage of laparoscopy in TP is histological confirmation [19]. Most cases of abdominal TB develop peritoneal adhesions, the peritoneum being in direct anatomical relations with the organs and the abdominal wall; thus, the presence of a common formation of fibrous bands and adhesions between the intestinal loops [21] with risk of occlusion is noted [22].

The TP treatment is largely similar to the TB treatment; therefore, a multi-drug protocol is used [23]. The standard four-drug regimen, consisting of rifampicin, isoniazid, pyrazinamide, and ethambutol, is recommended for PB [24]. These four drugs are used three times a week for the first two months, followed by isoniazid and rifampin for another four months. This protocol is usually effective, with good cure rates. When laboratory tests indicate resistance to first-line anti-tuberculous drugs, appropriate changes should be made [5].

The combination of corticosteroid therapy for a variable period of 1–3 months can prevent complications, especially those related to the formation of adhesions and the onset of sub-occlusive syndromes, as supported by many researchers; however, there is a controversy about the benefits associated with this therapy because steroids appear to have some benefits in patients with BP, but the poor quality of the studies limits the generalization of the results, and their use is not currently recommended [25].

It is necessary to wait for prospective, well-controlled, long-term clinical studies to be able to identify the optimal category of patients to benefit from such therapy. Based on the limited data available in the literature, it is currently difficult to make firm recommendations regarding the use of corticosteroids in TB patients [26].

Immunosuppression related to chronic kidney disease increases the risk of infection, bacillary peritonitis being, in this case, a rare cause of exacerbation of renal dysfunction that can lead to increased patient morbidity and mortality, in accordance with the cohort study carried out by Kao-Chi et al., where the overall incidence of TB was found to be 1.47-times higher in the CKD group compared to the non-CKD group [27].

Current anti-tuberculostatic treatments vary in duration, efficacy, safety, and tolerability. Treatment success rates fall below the recommended WHO threshold of 85%, leading to increased drug resistance. Antibiotics approved for non-TB bacterial infections show promise against multidrug-resistant TB [10].

Recent years have witnessed the growing utilization of advanced biomedical technologies to restore tissue and organ function. Notably, mesenchymal stem cell therapy holds potential for tissue repair. These versatile cells possess antimicrobial properties, migration ability to injury sites, and secrete bioactive molecules that aid in tissue restoration. Muraviov A.N. et al. conducted an effective experimental study, combining stem cell administration with standard anti-tuberculostatic therapy in rabbits, showcasing promise [28].

Another therapeutic achievement in the future can be represented by nanomedicine, with the advantage of using liposomes as carriers of antibiotics ranging from reducing toxicity to improving pharmacokinetic parameters and, in particular, biodistribution [29].

A TB granuloma is a complex cellular construct consisting of Mycobacterium tuberculosis-infected macrophages surrounded by other immune system cells, fatty acids, and cholesterol. It is very difficult to administer anti-TB drugs at the site of TB infection, as it has very poor vascularity, so nanotechnology may offer a solution to this problem [26,30]. Nanotechnology is traditionally used in immunotherapy and oncology but could also represent a revolution in the administration of antibiotics, where much remains to be discovered [29].

Abdominal complications were present in our patient who presented with intestinal obstruction, and according to the specialized literature, approximately 20–40% of patients with Tuberculosis Peritonitis (TP) develop an acute abdomen and require surgical treatment [31].

Intestinal obstruction is one of the most devastating complications of TP, necessitating emergency surgical intervention. Generally, patients who do not develop intestinal obstruction have a better prognosis [32].

## 4. Conclusions

In conclusion, TP has nonspecific symptoms, and its diagnosis requires a high degree of suspicion. Patients who have neutrophil-predominant peritoneal fluid and concomitant bacterial peritonitis are uncommon. Thus, peritoneal TB should be considered in patients with culture-negative peritonitis and in patients with culture-positive peritonitis unresponsive to appropriate antibiotics. Laparoscopy with biopsy may be considered at an early stage when TB peritonitis is suspected.

Non-invasive tests are usually insufficient; therefore, the sovereign diagnosis of bacillary peritonitis is exploratory laparoscopy with peritoneal biopsy, but there is also the risk of contamination related to any instrumental maneuver.

TP is a rare form of tuberculosis that affects the peritoneum, the lining of the abdominal cavity. It is difficult to diagnose because the symptoms are similar to other diseases, and it requires a combination of clinical, radiological, and biochemical tests to confirm it.

Overall, this is a challenging disease to diagnose and treat, and early recognition and appropriate management are necessary for a good outcome.

## Figures and Tables

**Figure 1 reports-06-00044-f001:**
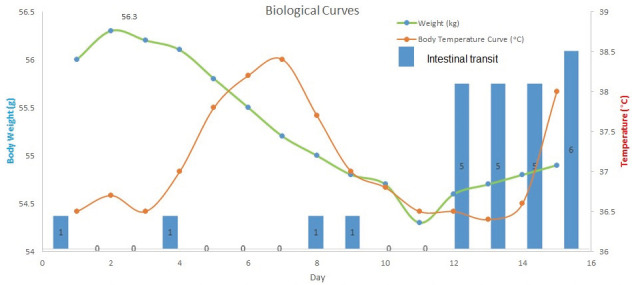
Biological curves. The presence of elevated body temperature signifies the dissemination of the infection. While under medical care in the hospital, the patient experienced a febrile episode, reaching a peak temperature of 38.5 degrees Celsius on the seventh day. Additionally, there was considerable fluctuation in bowel movements throughout the course of the illness, characterized by sporadic constipation from days 1 to 12, transitioning to severe diarrhea from days 12 to 16.

**Figure 2 reports-06-00044-f002:**
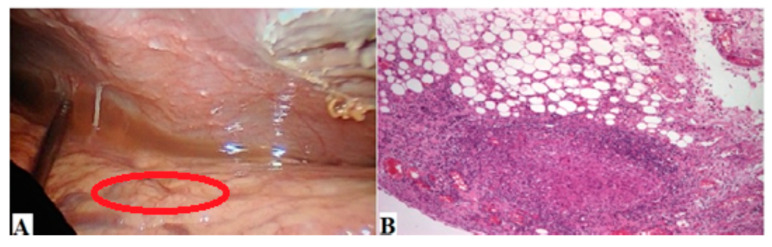
(**A**) Intraperitoneal laparoscopy revealed nodules with the presence of ascites fluid. The macroscopic aspect highlights multiple parietal and visceral peritoneal nodules (red circle), cloudy-looking ascites fluid in the splenic lodge, intra-hepato-diaphragmatic and Douglas sac. (**B**) Histochemical staining with prepared hematoxylin and eosin showed an image of epithelial granuloma forming Langhans cells (×10).

**Figure 3 reports-06-00044-f003:**
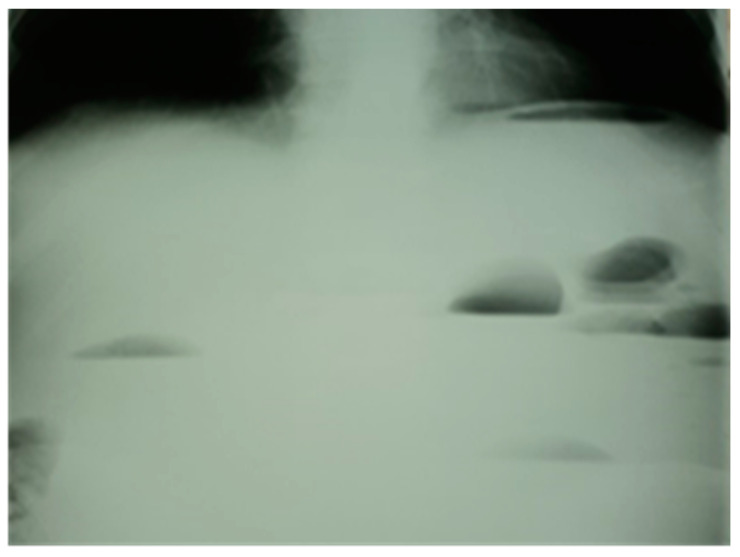
Simple abdominal X-ray, important aerocolia.

**Table 1 reports-06-00044-t001:** Biological parameters.

Biological Parameters	Normal Value	Value at Hospitalization	Evolving Values
CBC			
-Hb	11–16 mg/dL	7.3	9.5
-Leukocytes	4000–10,000/mm^3^	6500	18,000
-Platelets	150,000–450,000/mm^3^	109,000	108,000
Kidney function			
-Creatinine	0.72–1.25 mg/dL	8	12
-Urea	18–55 mg/dL	286	204
eGFR	90–120 mL/min/1.73 m^2^	7	4
Acid-base balance			
Alkaline reserve	22–30 mEq/L	7	12
Disorders of phospho-calcium metabolism			
-Serum calcium	8.4–10.2 mg/dL	5.1	6.7
-Phosphor	2.3–4.7 mg/dL	9.2	
-IPTH	15–65 pg/mL	248	
Inflammatory tests			
-Fibrinogen	200–400 mg/dL	497	
-C Reactive Protein	0–5 mg/L	12	
Urine summary examination			
-Density	1.005–1.025	1.004	
-Glucose	N ≤ 50 mg/dL)	500	
Virological profile			
HIV-1 RNA	neg	neg	
Atg Hbs	neg	neg	
Atc HCV	neg	neg	

**Table 2 reports-06-00044-t002:** Biological parameters: ascites fluid.

-Ascites fluid protein (g/dL)	5.1
-Glucose (N 90–110 mg/dL)	85
-Citological examination	frequent lymphocytes, leukocytes, PMN, Neutophils
-Microbiological examination	*Klebsiella*, Enterobacter
-B.K exam (Ziehl-Neelsen stain)	Negative
-ADA (0–9) U/L^6^	42 U/L^6^

## Data Availability

The data presented in this case report are available on request from the corresponding author.

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
