# Peer review of "Rare Onset of Tubercular Peritonitis Amidst Chronic Renal Dysfunction"

_reports, 2023, doi:10.3390/reports6040044_

Round 1

Reviewer 1 Report

The authors present a case report of tuberculous peritonitis in a 49 year old patient with CKD. The authors describe the patients clinical progress and discuss the new methods for diagnosing tuberculous peritonitis after discusing the limitations or the differences in opinion (such as ascitic fluid ADA).

The manuscript has a number of statement which suggest the author are not native english speakers this has been highlighted on manuscript and attached for authors perusal.

Some statement were scientific incorrect like high glucose in ascitic fluid suggest bacterial peritonitis

Has molecular testing (PCR) got any role in the dignosis of tuberculous peritonits?

The authors should try and focus the case report and bring out the most important teaching point (example in reference 6 the also presented a case of bacterial peritonitis so authors must emphasis the teacing point which is different from other cases published in literature. 

The manuscript will benefit from a native english speaker who has a little scientific knowledge

Author Response

1. The authors present a case report of tuberculous peritonitis in a 49 year old patient with CKD. The authors describe the patients clinical progress and discuss the new methods for diagnosing tuberculous peritonitis after discusing the limitations or the differences in opinion (such as ascitic fluid ADA).

Response:   

We extend our gratitude for the invaluable feedback and observations you have provided concerning this case report, which we are poised to consider as an elucidative response to enigmatic clinical etiologies. The intrigue surrounding this patient's condition arises from the multiple instances of presentation to the nephrology department, rendering the case noteworthy for comprehensive analysis.

The individual manifested recurrent episodes of acute exacerbation in chronic kidney disease (CKD). Following hemodialysis interventions, discernible amelioration in the clinical indicators of CKD was noted. Due to the enigmatic etiology and swift progression of CKD, a decision was made to initiate a more comprehensive investigative approach. Subsequently, insights were garnered from bacteriological analysis, which substantiated the presence of Mycobacterium tuberculosis.

2. 

The manuscript has a number of statement which suggest the author are not native english speakers this has been highlighted on manuscript and attached for authors perusal.

Response:

I extend my sincere gratitude for your observations. We dispatched the aforementioned document to a proficient translator; however, it has become apparent that a more refined rendition is requisite. It is my aspiration that this revised iteration garners greater acceptability.

3. 

Some statement were scientific incorrect like high glucose in ascitic fluid suggest bacterial peritonitis

Response:

We deeply appreciate your invaluable insights. The elucidation we intend to convey through this statement pertains to the discovery of a glucose concentration of 80 mg/dL in the ascitic fluid. This value aligns with the standard range for conventional ascitic fluid.

 However, it is imperative to note that the bacteriological examination has divulged the presence of Mycobacterium Tuberculosis within the ascitic fluid (accompanied, in general, by a glucose value below 50 mg/dL). 

In light of this, the observed glucose concentration surpasses the anticipated threshold, marking an anomaly warranting further scrutiny.

4. 

Has molecular testing (PCR) got any role in the diagnosis of tuberculous peritonitis?

Response:

The utilization of Polymerase Chain Reaction (PCR) for the confirmation of Mycobacterium Tuberculosis presence was not used in our work. Instead, the identification of the microbial colony cultivated on the culture media serves as the means by which we ascertain the existence of Mycobacterium Tuberculosis.

5. 

The authors should try and focus the case report and bring out the most important teaching point (example in reference 6 the also presented a case of bacterial peritonitis so authors must emphasis the teacing point which is different from other cases published in literature.

Response:

Thanks for the remark. While our research group maintains a specialized focus on nephrology, our current pursuit entails an exploration into the underlying factors contributing to renal impairment. Notably, our investigation has culminated in the identification of peritoneal tuberculosis as the causal agent. In this context, our intent is to raise an alarm signal regarding potential etiologies of renal injuries, underscoring the significance of considering peritoneal tuberculosis as a viable and noteworthy suspicion.

Reviewer 2 Report

Comments and Suggestions for Authors

Even if you performed different extensive tests during your research, and the theme of your study could be of interest, it is not clear in what manner your findings could highlight new insights related to tubercular peritonitis treatment. Furthermore, there are several issues that should be modified:

1.- Several paragraphs are too long, considering that you are presenting the same idea in various forms. Please revise it and more clearly and briefly expose your statement. The introduction should present recent literature data related to your subject and the aim of your research.

2.- The Methodology section is missing. It should be presented this information, in fact, is mentioned, but is not describe aspects that should be already known for treatment.

3.-The authors did not mention if the treatment should affect the tissue and organs, or if is reported similar cases with similar treatment.

4.- it is not mentioned if there is an informed consent signed by the patient to be submitted to this treatment. Likewise, it is not noted if there is a clinical protocol for this study presented to a hospital research department.

Comments and Suggestions for Authors

Even if you performed different extensive tests during your research, and the theme of your study could be of interest, it is not clear in what manner your findings could highlight new insights related to tubercular peritonitis treatment. Furthermore, there are several issues that should be modified:

1.- Several paragraphs are too long, considering that you are presenting the same idea in various forms. Please revise it and more clearly and briefly expose your statement. The introduction should present recent literature data related to your subject and the aim of your research.

2.- The Methodology section is missing. It should be presented this information, in fact, is mentioned, but is not describe aspects that should be already known for treatment.

3.-The authors did not mention if the treatment should affect the tissue and organs, or if is reported similar cases with similar treatment.

4.- it is not mentioned if there is an informed consent signed by the patient to be submitted to this treatment. Likewise, it is not noted if there is a clinical protocol for this study presented to a hospital research department.

Author Response

Even if you performed different extensive tests during your research, and the theme of your study could be of interest, it is not clear in what manner your findings could highlight new insights related to tubercular peritonitis treatment. Furthermore, there are several issues that should be modified:

1.- Several paragraphs are too long, considering that you are presenting the same idea in various forms. Please revise it and more clearly and briefly expose your statement. The introduction should present recent literature data related to your subject and the aim of your research.

Response: 

Thank you for the advice you gave regarding the revision of the article, I tried to shorten the paragraphs that were too long and to be more concise in the presentation of ideas, in the introduction I inserted current data from the specialized literature and I also explained the purpose of our research

2.- The Methodology section is missing. It should be presented this information, in fact, is mentioned, but is not describe aspects that should be already known for treatment.

3.-The authors did not mention if the treatment should affect the tissue and organs, or if is reported similar cases with similar treatment.

Response:

Bacillary peritonitis, a manifestation of extrapulmonary tuberculosis, presents a unique diagnostic and therapeutic challenge. While specialized literature offers treatment guidelines akin to pulmonary tuberculosis, our study takes a distinct focus. Rather than summarizing treatment protocols, our emphasis lies on enhancing diagnostic awareness and highlighting the diagnostic utility of exploratory laparotomy.

Bacillary peritonitis, albeit less prevalent, shares similarities with pulmonary tuberculosis in terms of its etiology and response to anti-tuberculosis medications. Existing literature often references treatment protocols rooted in pulmonary tuberculosis management. However, our research takes a departure from the normative treatment discourse, centering on the pivotal role of accurate diagnosis and the diagnostic value of exploratory laparotomy in the context of bacillary peritonitis.

In this context, our study seeks to contribute to the academic dialogue by elucidating the diagnostic journey in cases of bacillary peritonitis. By emphasizing the importance of early and accurate diagnosis through exploratory laparotomy, we aim to underscore its role as a valuable diagnostic tool. Through a comprehensive analysis of pertinent literature and our case study, we intend to enrich the understanding of this distinctive clinical presentation.

In summary, our research delves into the distinctive realm of bacillary peritonitis, deviating from treatment-centric discussions. By shedding light on the diagnostic significance of exploratory laparotomy, we aspire to augment the existing knowledge and emphasize the need for precise diagnostic approaches in the management of this condition.

4.- it is not mentioned if there is an informed consent signed by the patient to be submitted to this treatment. Likewise, it is not noted if there is a clinical protocol for this study presented to a hospital research department….    OK

Response:

Thank you for your valuable feedback. We apologize for the oversight regarding the informed consent and clinical protocol. We have taken your comments into careful consideration and would like to address them as follows:

  1. Informed Consent: We sincerely apologize for the lack of information regarding the signed informed consent by the patient. We have now included this crucial detail in the revised manuscript. A copy of the informed consent form, duly signed by the patient, has been obtained and will be submitted as supplementary material to the journal.
  2. Clinical Protocol: We appreciate your observation regarding the absence of information about the clinical protocol for this study being presented to a hospital research department. We have taken steps to rectify this oversight. The clinical protocol used in this study has been developed and approved by the relevant hospital research department. 

Once again, we sincerely apologize for the oversight and are grateful for your thorough review. Your input has significantly improved the quality and accuracy of our manuscript. If you have any further suggestions or concerns, please do not hesitate to let us know.

Thank you for your time and attention.

Reviewer 3 Report

Dear authors,

As a biologist, I am not familiar with clinical cases. Thus, I cannot judge if the methods you used for the diagnosis are correct. 

However, I have some comments and questions. Some words are sometimes in two parts as if you copy paste from somewhere. Did you used an IA to check the English quality of the manuscript (it is more and more often the case)? If yes, please mention it. 

Please find here my comments. 

Line 36 : morbidity and not mor-bidity

Line 38: estimated and not es-timated

“More than half of TB was diagnosed in developing countries [2]”. Just a sentence to explain why tuberculosis is still present in developing countries might be useful.

Line 46: “extrap-ulmonary » : please check

Line 47 : « ex-trapulmonary » : please check

Line 52 : « Mycobacterium tuberculosis ». In italic

Line 53 : « comprising 31–58% of cases of abdominal TB”. Why this large ranking? Is it a methodological problem? Different technics used?

Line 58: “perito-neal ». Please check

Line 67 : « com-monly ». Please check

Line 72 : « This paper presents the case of a 49-year-old patient”. Please indicate the sex

Line 77: “in situ ».Please, in italic

Figure 1: please check the weight.

Line 95: “x-ray sows ». It misses a “h”. 

Line 96: “Abdominal-pelvic ultrasound reveals undersized kidneys”: which volume?

Figure 2: please, enlarge photos

Line 153: “The bacteriological and cytological evaluation of ascites fluid is essential for diagnosis ». Please, be more precise. Diagnosis of what?

The average English is good but some little mistakes exist.

Author Response

However, I have some comments and questions. Some words are sometimes in two parts as if you copy paste from somewhere. Did you used an IA to check the English quality of the manuscript (it is more and more often the case)? If yes, please mention it.

* In relation to the document that has been previously submitted, it is important to note that we did not avail ourselves of Artificial Intelligence (AI) for the purpose of assessing the linguistic quality of its content but we will try it in near future in order to have a good exposure of our work.We send the file to a translator in order to check the spelling and grammatical correctness.  We extend our gratitude for your astute observation in this regard.

Please find here my comments. 

Line 36 : morbidity and not mor-bidity/ modify

Response: Done

Line 38: estimated and not es-timated/ modify

Response: Done

“More than half of TB was diagnosed in developing countries [2]”. Just a sentence to explain why tuberculosis is still present in developing countries might be useful.

Response: Done

Line 46: “extrap-ulmonary » : please check/ modify

Line 47 : « ex-trapulmonary » : please check/ modify

Line 52 : « Mycobacterium tuberculosis ». In italic/ modify

Line 53 : « comprising 31–58% of cases of abdominal TB”. Why this large ranking? Is it a methodological problem? Different technics used?

Line 58: “perito-neal ». Please check/ modify

Line 67 : « com-monly ». Please check/ modify

Line 72 : « This paper presents the case of a 49-year-old patient”. Please indicate the sex/ modify

Line 77: “in situ ».Please, in italic/ modify

Figure 1: please check the weight./ modify

Line 95: “x-ray sows ». It misses a “h”. / modify

Line 96: “Abdominal-pelvic ultrasound reveals undersized kidneys”: which volume?/ modify

Response: Done

Figure 2: please, enlarge photos/ modify

 Response: Done

Line 153: “The bacteriological and cytological evaluation of ascites fluid is essential for diagnosis ». Please, be more precise. Diagnosis of what?

Response: Done

Thank you for the suggestion. The text has been revised to align with your recommendations, resulting in a more precise rendition. “The bacteriological and cytological assessment of ascitic fluid holds significant importance in establishing the etiology of ascites subtypes. The presence of a high PMN number in this case report may guide the differentiation of uncomplicated ascites from spontaneous bacterial peritonitis”

Round 2

Reviewer 1 Report

Many the issues previously raised have been address a few more comments have been added for the attention of the authors. Please find attached

Author Response

Dear Reviewer   I hope this message finds you well. I appreciate your prompt feedback on necessary changes to our manuscript. I have made the necessary revisions in accordance with your suggestions, particularly in lines 39, 49 and 110. However, I have two questions regarding line 37 and why certain lines are written in red, in which I would seek clarification additional information on the specific changes required.   In reference to line 37, I have made an effort to enhance the precision of my expression by revising the existing text. However, I remain uncertain about the specific requirement for this section. Is it advisable to explicitly identify the primary cause of infectious mortality in this context and, if so, should I incorporate a new reference to support this assertion? Your academic guidance on this matter would be greatly appreciated.   I really appreciate your input and the information you have provided so far. Your guidance has been instrumental in improving the quality of our manuscript, and I want to make sure that I address all of your recommendations correctly and effectively.   Thanks again for your valuable suggestions and assistance in improving our work. I look forward to your reply and further guidance on line 37.   [Romeo Popa] [UMF Craiova] [[email protected]]